# Scalable Model Selection for Belief Networks

**Zhao Song**[†]**, Yusuke Muraoka**[∗]**, Ryohei Fujimaki**[∗]**, Lawrence Carin**[†]

[†]Department of ECE, Duke University
Durham, NC 27708, USA
{zhao.song, lcarin}@duke.edu

[∗]NEC Data Science Research Laboratories
Cupertino, CA 95014, USA
{ymuraoka, rfujimaki}@nec-labs.com

## Abstract

We propose a scalable algorithm for model selection in sigmoid belief networks (SBNs), based on the factorized asymptotic Bayesian (FAB) framework. We derive the corresponding generalized factorized information criterion (gFIC) for the SBN, which is proven to be statistically consistent with the marginal log-likelihood. To capture the dependencies within hidden variables in SBNs, a recognition network is employed to model the variational distribution. The resulting algorithm, which we call FABIA, can simultaneously execute both model selection and inference by maximizing the lower bound of gFIC. On both synthetic and real data, our experiments suggest that FABIA, when compared to state-of-the-art algorithms for learning SBNs, $(i)$ produces a more concise model, thus enabling faster testing; $(ii)$ improves predictive performance; $(iii)$ accelerates convergence; and $(iv)$ prevents overfitting.

## 1 Introduction

The past decade has witnessed a dramatic increase in popularity of deep learning [20], stemming from its state-of-the-art performance across many domains, including computer vision [19], reinforcement learning [27], and speech recognition [15]. However, one important issue in deep learning is that its performance is largely determined by the underlying model: a larger and deeper network tends to possess more representational power, but at the cost of being more prone to overfitting [32], and increased computation. The latter issue presents a challenge for deployment to devices with constrained resources [2]. Inevitably, an appropriate model-selection method is required to achieve good performance. Model selection is here the task of selecting the number of layers and the number of nodes in each layer.

Despite the rapid advancement in performance of deep models, little work has been done to address the problem of model selection. As a basic approach, cross-validation selects a model according to a validation score. However, this is not scalable, as its complexity is exponential with respect to the number of layers in the network: $\mathcal{O}(J_{\text{MAX}}^{L_{\text{MAX}}})$, where $J_{\text{MAX}}$ and $L_{\text{MAX}}$ represent the maximum allowed numbers of nodes in each layer and number of layers, respectively. In Alvarez and Salzmann [2], a constrained optimization approach was proposed to infer the number of nodes in convolutional neural networks (CNNs); the key idea is to incorporate a sparse group Lasso penalty term to shrink all edges flowing into a node. Based on the shrinkage mechanism of the truncated gamma-negative binomial process, Zhou et al. [36] showed that the number of nodes in Poisson gamma belief networks (PGBNs) can be learned. Furthermore, we empirically observe that the shrinkage priors employed in Gan et al. [11], Henao et al. [14], Song et al. [31] can potentially perform model selection in certain tasks, even though this was not explicitly discussed in those works. One common problem for these approaches, however, is that the hyperparameters need to be tuned in order to achieve good performance, which may be time-consuming for some applications involving deep networks.

The factorized asymptotic Bayesian (FAB) approach has recently been shown as a scalable model-selection framework for latent variable models. Originally proposed for mixture models [9], it was later extended to the hidden Markov model (HMM) [8], latent feature model (LFM) [12], and relational model [22]. By maximizing the approximate marginal log-likelihood, FAB introduces an $\ell_0$ regularization term on latent variables, which can automatically estimate the model structure by eliminating irrelevant latent features through an expectation maximization [7] (EM)-like alternating optimization, with low computational cost.

We develop here a scalable model selection algorithm within the FAB framework to infer the size of SBNs [28], a popular component of deep models, e.g., deep belief networks (DBN) [16] and deep Poisson factor analysis (DPFA) [10], and we assume here the depth of the SBN is fixed. Since the mean-field assumption used in FAB does not hold in SBNs, we employ a recognition network [18, 29, 25, 26] to represent the variational distribution. As our method combines the advantages of FAB Inference and Auto-encoding variational Bayesian (VB) frameworks, we term it as FABIA. To handle large datasets, we also derive a scalable version of FABIA with mini-batches. As opposed to previous works, which predefine the SBN size [28, 30, 25, 5, 11, 6, 31, 26], FABIA determines it automatically.

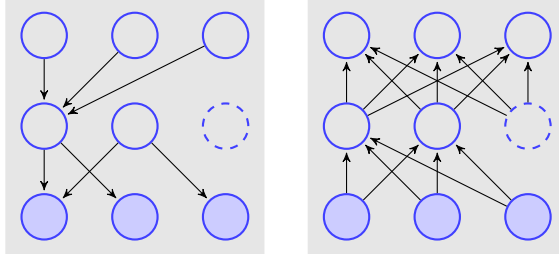

Figure 1: Requirement for removal of nodes in (Left) SBN and (Right) FNN (dashed circles denote nodes that can be removed). Note that a node in the SBN can be removed only if all of its connected edges shrink. For FNN, shrinkage of all incoming edges eliminates a node.

It should be noted that model selection in SBNs is more challenging than CNNs and feedforward neural networks (FNNs). As shown in Figure 1, simply imposing a sparsity prior or a group sparsity prior as employed in CNNs [2] and SBNs [11, 14, 31] does not necessarily shrink a node in SBN, since such approaches cannot guarantee to shrink all edges connected to a node.

FABIA possesses the following distinguishing features: $(i)$ a theoretical guarantee that its objective function, the generalized factorized information criterion (gFIC), is statistically consistent with the model's marginal log-likelihood; and $(ii)$ prevention of overfitting in large networks when the amount of training data is not sufficiently large, thanks to an intrinsic shrinkage mechanism. We also detail that FABIA has important connections with previous work on model regularization, such as Dropout [32], Dropconnect [35], shrinkage priors [11, 36, 14, 31], and automatic relevance determination (ARD) [34].

## 2 Background

An SBN is a directed graphical model for which the distribution of each layer is determined by the preceding layer via the sigmoid function, defined as $\sigma(x) \triangleq 1/[1 + \exp(-x)]$. Let $\boldsymbol{h}^{(l)}$ denote the $l$th hidden layer with $J_l$ units, and $\boldsymbol{v}$ represent the visible layer with $M$ units. The generative model of the SBN, with $L$ hidden layers, is represented as

$$p(\boldsymbol{h}^{(L)}|\boldsymbol{b}) = \prod_{i=1}^{J_L}[\sigma(b_i)]^{h_i^{(L)}}\,[\sigma(-b_i)]^{1-h_i^{(L)}}, \qquad p(\boldsymbol{h}^{(l)}|\boldsymbol{h}^{(l+1)}) = \prod_{i=1}^{J_l}[\sigma(\psi_i^{(l)})]^{h_i^{(l)}}\,[\sigma(-\psi_i^{(l)})]^{1-h_i^{(l)}}$$

where $l = 1, \ldots, L-1$, $\psi_i^{(l)} = W_{i\cdot}^{(l)}\boldsymbol{h}^{(l+1)} + c_i^{(l)}$, and $\boldsymbol{b}$ corresponds to prior parameters; the notation $i\cdot$ means the $i$th row of a matrix. For the link function of the visible layer, i.e., $p(\boldsymbol{v}|\boldsymbol{h}^{(1)})$, we use the sigmoid function for binary data and the multinomial function for count data, as in Mnih and Gregor [25], Carlson et al. [6].

One difficulty of learning SBNs is the evaluation of the expectation with respect to the posterior distribution of hidden variables [31]. In Mnih and Gregor [25], a recognition network under the variational auto-encoding (VAE) framework [18] was proposed to approximate this intractable expectation. Compared with the Gibbs sampler employed in Gan et al. [11], Carlson et al. [6], Song et al. [31], the recognition network enables fast sampling of hidden variables in blocks. The variational parameters in

the recognition network can be learned via stochastic gradient descent (SGD), as shown in the neural variational inference and learning (NVIL) algorithm [25], for which multiple variance reduction techniques have been proposed to obtain better gradient estimates. Note that all previous work on learning SBNs assumes that a model with a fixed number of nodes in each layer has been provided.

To select a model for an SBN, we follow the FAB framework [9], which infers the structure of a latent variable model by Bayesian inference. Let $\boldsymbol{\theta} = \{W, \boldsymbol{b}, \boldsymbol{c}\}$ denote the model parameters and $\mathcal{M}$ be the model, with the goal in the FAB framework being to obtain the following maximum-likelihood (ML) estimate:

$$\widehat{\mathcal{M}}_{\text{ML}} \;=\; \arg\max_{\mathcal{M}} \sum_{n=1}^{N} \ln p(\boldsymbol{v}_n|\mathcal{M}) \;=\; \arg\max_{\mathcal{M}} \sum_{n=1}^{N} \ln \sum_{\boldsymbol{h}_n} \int p(\boldsymbol{v}_n, \boldsymbol{h}_n|\boldsymbol{\theta}) p(\boldsymbol{\theta}|\mathcal{M}) \, d\boldsymbol{\theta} \qquad (1)$$

As a key feature of the FAB framework, the $\ell_0$ penalty term on $\boldsymbol{h}_n$ induced by approximating (1) can remove irrelevant latent variables from the model ("shrinkage mechanism"). In practice, we can start from a large model and gradually reduce its size through this "shrinkage mechanism" until convergence.

Although a larger model has more representational capacity, a smaller model with similar predictive performance is preferred in practice, given a computational budget. A smaller model also enables faster testing, a desirable property in many machine learning tasks. Furthermore, a smaller model implies more robustness to overfitting, a common danger in deeper and larger models with insufficient training data.

Since the integration in (1) is in general intractable, Laplace's method [23] is employed in FAB inference for approximation. Consequently, gFIC can be derived as a surrogate function of the marginal log-likelihood. By maximizing the variational lower bound of gFIC, one obtains estimates of both parameters and the underlying model size. Note that while FAB inference uses the mean-field approximation for the variational distribution [9, 8, 22, 21], the same does not hold for SBNs, due to the correlation within hidden variables given the data. In contrast, the recognition network has been designed to approximate the posterior distribution of hidden variables with more fidelity [18, 29, 25]. Therefore, it can be a better candidate for the variational distribution in our task.

## 3 The FABIA Algorithm

### 3.1 gFIC for SBN

Following the FAB inference approach, we first lower bound the marginal log-likelihood in (1) via a variational distribution $q(\boldsymbol{h}|\boldsymbol{\phi})$ as [1]

$$\ln \sum_{\boldsymbol{h}_n} \int p(\boldsymbol{v}_n, \boldsymbol{h}_n|\boldsymbol{\theta}) p(\boldsymbol{\theta}|\mathcal{M}) \, d\boldsymbol{\theta} \;\geq\; \sum_{\boldsymbol{h}_n} q(\boldsymbol{h}_n|\boldsymbol{\phi}) \ln \left[ \frac{\int p(\boldsymbol{v}_n, \boldsymbol{h}_n|\boldsymbol{\theta}) \, p(\boldsymbol{\theta}|\mathcal{M}) \, d\boldsymbol{\theta}}{q(\boldsymbol{h}_n|\boldsymbol{\phi})} \right].$$

By applying Laplace's method [23], we obtain

$$\ln p(\boldsymbol{v}, \boldsymbol{h}|\mathcal{M}) = \frac{D_{\boldsymbol{\theta}}}{2} \ln(\frac{2\pi}{N}) + \sum_{n=1}^{N} \ln p(\boldsymbol{v}_n, \boldsymbol{h}_n|\widehat{\boldsymbol{\theta}}) + \ln p(\widehat{\boldsymbol{\theta}}|\mathcal{M}) - \frac{1}{2} \sum_{m=1}^{M} \ln |\Psi^m| + \mathcal{O}(1) \quad (2)$$

where $D_{\boldsymbol{\theta}}$ refers to the dimension of $\boldsymbol{\theta}$, $\widehat{\boldsymbol{\theta}}$ represents the ML estimate of $\boldsymbol{\theta}$, and $\Psi^m$ represents the negative Hessian of the log-likelihood with respect to $W_{m\cdot}$.

Since $\ln |\Psi^m|$ in (2) cannot be represented with an analytical form, we must approximate it first, for the purpose of efficient optimization of the marginal log-likelihood. Following the gFIC [13] approach, we propose performing model selection in SBNs by introducing the shrinkage mechanism from this approximation. We start by providing the following assumptions, which are useful in the proof of our main theoretical results in Theorem 1.

**Assumption 1.** *The matrix $\sum_{n=1}^{N} \eta_n \boldsymbol{h}_n^T \boldsymbol{h}_n$ has full rank with probability 1 as $N \to \infty$, where $\eta_n \in (0, 1)$.*

Note that this full-rank assumption implies that the SBN can preserve information in the large-sample limit, based on the degeneration analysis of gFIC [13].

**Assumption 2.** $h_{n,j}, \forall j$ is generated from a Bernoulli distribution as $h_{n,j} \sim Ber(\tau_j)$, where $\tau_j > 0$.

**Theorem 1.** As $N \to \infty$, $\ln |\Psi^m|$ can be represented with the following equality:

$$\ln |\Psi^m| = \sum_j \Big( \ln \sum_n h_{n,j} - \ln N \Big) + \mathcal{O}(1) \tag{3}$$

*Proof.* We first compute the negative Hessian as

$$\Psi^m = -\frac{1}{N} \frac{\partial}{\partial W_{m\cdot}^T \, \partial W_{m\cdot}} \sum_n \ln p(\boldsymbol{v}_n, \boldsymbol{h}_n | \boldsymbol{\theta}) = \frac{1}{N} \sum_n \sigma(W_m \cdot \boldsymbol{h}_n) \, \sigma(-W_m \cdot \boldsymbol{h}_n) \, \boldsymbol{h}_n^T \, \boldsymbol{h}_n.$$

From Assumption 1, $\Psi^m$ has full rank, since $\sigma(x) \in (0,1), \forall x \in \mathbb{R}$. Furthermore, the determinant of $\Psi^m$ is bounded, since $\Psi_{ij}^m \in (0,1), \forall i,j$. Next, we define the following diagonal matrix

$$\Lambda \triangleq \text{diag} \left[ \left( \frac{(\sum_n h_{n,1})}{N} \right), \ldots, \left( \frac{(\sum_n h_{n,J})}{N} \right) \right].$$

From Assumption 2, $\lim_{N \to \infty} Pr[\sum_n h_{n,j} = 0] = 0, \forall j$. Therefore, $\Lambda$ is full-rank and its determinant is bounded, when $N \to \infty$. Subsequently, we can decompose it as

$$\Psi^m = \Lambda \, F \tag{4}$$

where $F$ also has full rank and bounded determinant. Finally, applying the log determinant operator to the right side of (4) leads to our conclusion. $\square$

To obtain the gFIC for SBN, we first follow the previous FAB approaches [9, 12, 22] to assume the log-prior of $\boldsymbol{\theta}$ to be constant with respect to $N$, i.e., $\lim_{N \to \infty} \frac{\ln p(\boldsymbol{\theta}|\mathcal{M})}{N} = 0$. We then apply Theorem 1 to (2) and have

$$\text{gFIC}_{\text{SBN}} = \max_q \mathbb{E}_q \left[ -\frac{M}{2} \sum_j \Big( \ln \sum_n h_{n,j} \Big) + \sum_{n=1}^{N} \ln p(\boldsymbol{v}_n, \boldsymbol{h}_n | \widehat{\boldsymbol{\theta}}) + \frac{MJ - D_{\boldsymbol{\theta}}}{2} \ln N \right] + H(q) \tag{5}$$

where $H(q)$ is the entropy for the variational distribution $q(\boldsymbol{h})$.

As a key quantity in (5), $\frac{M}{2} \sum_j (\ln \sum_n h_{n,j})$ can be viewed as a regularizer over the model to execute model selection. This term directly operates on hidden nodes to perform shrinkage, which distinguishes our approach from previous work [11, 14, 31], where sparsity priors are assigned over edges. As illustrated in Figure 1, these earlier approaches do not necessarily shrink hidden nodes, as setting up a prior or a penalty term to shrink all edges connected to a node is very challenging in SBNs. Furthermore, the introduction of this quantity does not bring any cost of tuning parameters with cross-validation. In contrast, the Lagrange parameter in Alvarez and Salzmann [2] and hyperparameters for priors in Gan et al. [11], Henao et al. [14], Zhou et al. [36], Song et al. [31] all need to be properly set, which may be time-consuming in certain applications involving deep and large networks.

Under the same regularity conditions as Hayashi and Fujimaki [12], gFIC$_{\text{SBN}}$ is statistically consistent with the marginal log-likelihood, an important property of the FAB framework.

**Corollary 1.** As $N \to \infty$, $\ln p(\boldsymbol{v}|\mathcal{M}) = \text{gFIC}_{\text{SBN}} + \mathcal{O}(1)$.

*Proof.* The conclusion holds as a direct extension of the consistency results in Hayashi and Fujimaki [12]. $\square$

## 3.2 Optimization of gFIC

The gFIC$_{\text{SBN}}$ in (5) cannot be directly optimized, because $(i)$ the ML estimator $\widehat{\boldsymbol{\theta}}$ is in general not available, and $(ii)$ evaluation of the expectation over hidden variables is computationally expensive. Instead, the proposed FABIA algorithm optimizes the lower bound as

$$\text{gFIC}_{\text{SBN}} \geq -\frac{M}{2} \sum_j \Big[ \ln \sum_n \mathbb{E}_q(h_{n,j}) \Big] + \sum_{n=1}^{N} \mathbb{E}_q \big[ \ln p(\boldsymbol{v}_n, \boldsymbol{h}_n | \boldsymbol{\theta}) \big] + H(q) \tag{6}$$

where we use the following facts to get the lower bound: $(i)$ $p(\boldsymbol{v}_n, \boldsymbol{h}_n | \widehat{\boldsymbol{\theta}}) \geq p(\boldsymbol{v}_n, \boldsymbol{h}_n | \boldsymbol{\theta}), \forall \boldsymbol{\theta}$; $(ii)$ the concavity of the logarithm function; $(iii)$ $D_{\boldsymbol{\theta}} \leq MJ$; and $(iv)$ the maximum of all possible variational distributions $q$ in (5).

This leaves the choice of the form of the variational distribution. We could use the mean-field approximation as in previous FAB approaches [9, 8, 12, 13, 22, 21]. However, this approximation fails to capture the dependencies between hidden variables in SBN, as discussed in Song et al. [31].

Instead, we follow the recent auto-encoding VB approach [18, 29, 25, 26] to model the variational distribution with a recognition network, which maps $\boldsymbol{v}_n$ to $q(\boldsymbol{h}_n | \boldsymbol{v}_n, \boldsymbol{\phi})$. Specifically, $q(\boldsymbol{h}_n | \boldsymbol{v}_n, \boldsymbol{\phi}) = \prod_{j=1}^{J} q(h_{n,j} | \boldsymbol{v}_n, \boldsymbol{\phi}) = \prod_{j=1}^{J} \text{Ber}[\sigma(\boldsymbol{\phi}_j \cdot \boldsymbol{v}_n)]$, where $\boldsymbol{\phi} \in \mathbb{R}^{J \times M}$ parameterizes the recognition network. Not only does using a recognition network allow us to more accurately model the variational distribution, it also enables faster sampling of hidden variables.

The optimization of the lower bound in (6) can be executed via SGD; we use the Adam algorithm [17] as our optimizer. To reduce gradient variance, we employ the NVIL algorithm to estimate gradients in both generative and recognition networks. We also note that other methods, such as the importance-sampled objectives method [5, 26, 24], can be used and such an extension is left for future work.

Since $\frac{M}{2} \sum_j \left[ \ln \sum_n \mathbb{E}_q(h_{n,j}) \right]$ in (6) is only dependent on $q$, gradients of the generative model in our FABIA algorithm and NVIL should be the same. However, gradients of the recognition network in FABIA are regularized to shrink the model, which is lacking in the standard VAE framework.

We note that FABIA is a flexible framework, as its shrinkage term can be combined with any gradient-based variational auto-encoding methods to perform model selection, where only minimal changes to the gradients of the recognition network of the original methods are necessary.

A node $j$ at level $l$ will be removed from the model if it satisfies $\frac{1}{N} \sum_{n=1}^{N} \mathbb{E}_q(h_{n,j}^{(l)}) \leq \epsilon^{(l)}$, where $\epsilon^{(l)}$ is a threshold parameter to control the model size. This criterion has an intuitive interpretation that a node should be removed if the proportion of its samples equaling 1 is small. When the expectation is not exact, such as in the top layers, we use samples drawn from the recognition network to approximate it.

### 3.3 Minibatch gFIC

To handle large datasets, we adapt the $\text{gFIC}_{\text{SBN}}$ developed in (5) to use minibatches (which is also appropriate for online learning). Suppose that each mini-batch contains $N_{mini}$ data points, and currently we have seen $T$ mini-batches, an unbiased estimator for (5) (up to constant terms) is then

$$\widetilde{\text{gFIC}_{\text{SBN}}} = \max_q \mathbb{E}_q \left[ -\frac{M}{2} \sum_j \ln \left( \sum_{i=1}^{N_{mini}} h_{i+N_T, j} \right) + T \sum_{i=1}^{N_{mini}} \ln \frac{p(\boldsymbol{v}_{i+N_T}, \boldsymbol{h}_{i+N_T} | \widehat{\boldsymbol{\theta}})}{q(\boldsymbol{h}_{i+N_T} | \boldsymbol{\phi})} \right.$$
$$\left. + \frac{MJ - D_{\boldsymbol{\theta}}}{2} \ln N_{T+1} \right] \tag{7}$$

where $N_T = (T-1) N_{mini}$. Derivation details are provided in Supplemental Materials.

An interesting observation in (7) is that $\widetilde{\text{gFIC}_{\text{SBN}}}$ can automatically adjust shrinkage over time: At the beginning of the optimization, i.e., when $T$ is small, the shrinkage term $\frac{M}{2} \sum_j \ln(\sum_{i=1}^{N_{mini}} h_{i+N_T, j})$ is more dominant in (7). As $T$ becomes larger, the model is more stable and shrinkage gradually disappears. This phenomenon is also observed in our experiments in Section 5.

### 3.4 Computational complexity

The NVIL algorithm has complexity $\mathcal{O}(MJN_{train})$ for computing gradients in both the generative model and recognition network. FABIA needs an extra model selection step, also with complexity $\mathcal{O}(MJN_{train})$ per step. As the number of training iteration increases, the additional cost to perform model selection is offset by the reduction of time when computing gradients, as observed in Figure 3. In test, the complexity is $\mathcal{O}(MJN_{test}K)$ per step, with $K$ being the number of samples taken to compute the variational lower bound. Therefore, shrinkage of nodes can linearly reduce the testing time.

## 4 Related Work

**Dropout**    As a standard approach to regularize deep models, Dropout [32] randomly removes a certain number of hidden units during training. Note that FABIA shares this important characteristic by directly operating on nodes, instead of edges, to regularize the model, which has a more direct connection with model selection. One important difference is that in each training iteration, Dropout updates only a subset of the model; in contrast, FABIA updates every parameter in the model, which enables faster convergence.

**Shrinkage prior**    The shrinkage sparsity-inducing approach aims to shrink edges in a model, by employing either shrinkage priors [11, 14, 36, 31] or a random mask [35] on the weight matrix. In FABIA, the penalty term derived in gFIC of (5) also has the shrinkage property, but the shrinkage effect is instead imposed on the nodes. Furthermore, shrinkage priors are usually approached from the Bayesian framework, where Markov chain Monte Carlo (MCMC) is often needed for inference. In contrast, FABIA integrates the shrinkage mechanism from gFIC into the auto-encoding VB approach and thus is scalable to large deep models.

**Group Sparsity**    Application of group sparsity can be viewed as an extension of the shrinkage prior, with the key idea being to enforce sparsity on entire rows (columns) of the weight matrix [2]. This corresponds to the ARD prior [34] where each row (column) has an individual hyperparameter. In FNNs and CNNs, this is equivalent to node shrinkage in FABIA for SBNs. The structure of SBNs precludes a direct application of the group sparsity approach for model selection, but there exists an interesting opportunity for future work to extend FABIA to FNNs and CNNs.

**Nonparametric Prior**    In Adams et al. [1], a cascading Indian buffet process (IBP) based approach was proposed to infer the structure of the Gaussian belief network with continuous hidden units, for which the inference was performed via MCMC. By employing the nonparametric properties of the IBP prior, this approach can adjust the model size with observations. Due to the high computational cost of MCMC, however, it may not be scalable to large problems.

## 5 Experiments

We test the proposed FABIA algorithm on synthetic data, as well as real image and count data. For comparison, we use the NVIL algorithm [25] as a baseline method, which does not have the model selection procedure. Both FABIA and NVIL are implemented in Theano [4] and tested on a machine with 3.0GHz CPU and 64GB RAM. The learning rate in Adam is set to be 0.001 and we follow the default settings of other parameters in all of our experiments. We set the threshold parameter $\epsilon^{(l)}$ to be 0.001, $\forall l$ unless otherwise stated. We also tested Dropout but did not notice any clear improvement. The purpose of these experiments is to show that FABIA can automatically learn the model size, and achieve better or competitive performance with a more compact model.

### 5.1 Synthetic Dataset

The synthetic data are generated from a one-layer SBN and a two-layer SBN, with $M = 30$ visible units in both cases. We simulate 1250 data points, and then follow an $80/20\%$ split to obtain the training and test sets. For the one-layer case, we employ a true model with 5 nodes and initialize FABIA and NVIL with 25 nodes. For the two-layer case, the true network has the structure of 10-5 [2], and we initialize FABIA and NVIL with a network of 25-15. We compare the inferred SBN structure and test log-likelihood for FABIA, the NVIL algorithm initialized with the same model size as FABIA (denoted as "NVIL"), and the NVIL algorithm initialized with the true model size (denoted as "NVIL (True)"). One hundred independent random trials are conducted to report statistics.

Figure 2(a) shows the mean and standard deviation of the number of nodes inferred by FABIA, as a function of iteration number. In both one- and two-layer cases, the mean of the inferred model size is very close to the ground truth. In Figure 2(b), we compare the convergence in terms of the test log-likelihood for different algorithms: FABIA has almost the same convergence speed as NVIL with

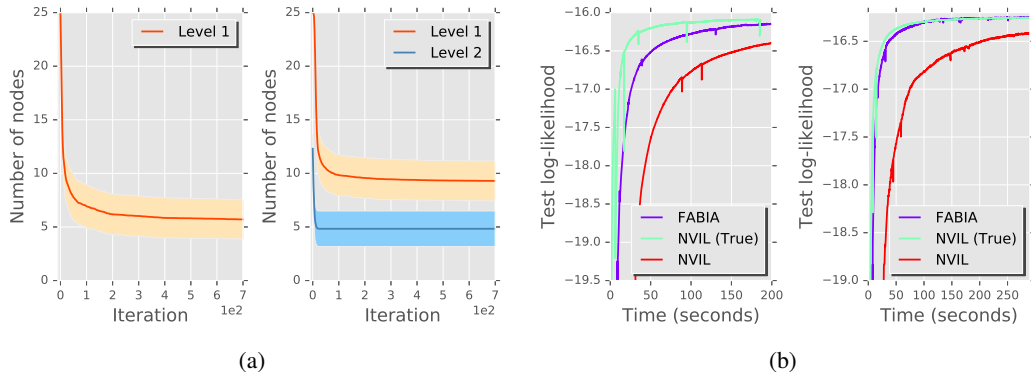

(a)                                                                          (b)

Figure 2: (a) Inferred number of nodes from FABIA in (Left) one- and (Right) two-layer cases; (b) Test log-likelihood for different methods in (Left) one- and (Right) two-layer cases.

the true model, both of which have remarkable gaps over the NVIL variant initialized with the same model size as FABIA.

## 5.2 Image Modeling

We use the publicly available MNIST dataset, which contains $60,000$ training and $10,000$ test images of size $28 \times 28$. Our performance metric is the variational lower bound of the test log-likelihood. The mini-batches for FABIA and NVIL are set to $100$. For this dataset we compared FABIA with the VB approach in Gan et al. [11] and Rec-MCEM in Song et al. [31]. The VB approach in Gan et al. [11] can potentially shrink nodes, due to the three parameter beta-normal (TPBN) prior [3]. We claim a node $h_j^{(l)}$ can be removed from the model, if its adjacent weight matrices satisfy $\sum_k [W_{k,j}^{(l)}]^2 / J^{(l-1)} < 10^{-8}$ and $\sum_k [W_{j,k}^{(l+1)}]^2 / J^{(l+1)} < 10^{-8}$. We run the code provided in `https://github.com/zhegan27/dsbn_aistats2015` and use default parameter settings to report the VB results. We also implemented the Rec-MCEM approach but only observed shrinkage of edges, not nodes.

Table 1: Model size, test variational lower bound (VLB) (in nats), and test time (in seconds) on the MNIST dataset. Note that FABIA and VB start from the same model size as NVIL and Rec-MCEM.

| Method | Size | VLB | Time |
|---|---|---|---|
| VB | 81 | -117.04 | 8.94 |
| Rec-MCEM | 200 | -116.70 | 8.52 |
| NVIL | 200 | -115.63 | 8.47 |
| FABIA | 107 | **−114.96** | **6.88** |
| VB | 200-11 | -113.69 | 22.37 |
| Rec-MCEM | 200-200 | -106.54 | 12.25 |
| NVIL | 200-200 | -105.62 | 12.34 |
| FABIA | 135-93 | **−104.92** | **9.18** |
| NVIL | 200-200-200 | -101.99 | 15.66 |
| FABIA | 136-77-72 | **−101.14** | **10.97** |

Table 1 shows the variational lower bound of the test log-likelihood, model size, and test time for different algorithms. FABIA achieves the highest test log-likelihood in all cases and converges to smaller models, compared to NVIL. FABIA also benefits from its more compact model to have the smallest test time. Furthermore, we observe that VB always over-shrinks nodes in the top layer, which might be related to the settings of hyperparameters. Unlike VB, FABIA avoids the difficult task of tuning hyperparameters to balance predictive performance and model size. We also notice that the deeper layer in the two-layer model did not shrink in VB, as our experiments suggest that all nodes in the deeper layer still have connections with nodes in adjacent layers.

Figure 3 shows the variational lower bound of the test log-likelihood and number of nodes in FABIA, as a function of CPU time, for different initial model sizes. Additional plots as a function of the number of iterations are provided in Supplemental Materials, which are similar to Figure 3. We note that FABIA initially has a similar log-likelihood that gradually outperforms NVIL, which can be explained by the fact that FABIA initially needs additional time to perform the shrinkage step but later converges to a smaller and better model. This gap becomes more obvious when we increase the number of hidden units from $200$ to $500$. The deteriorating performance of NVIL is most likely due to overfitting. In contrast, FABIA is robust to the change of the initial model size.

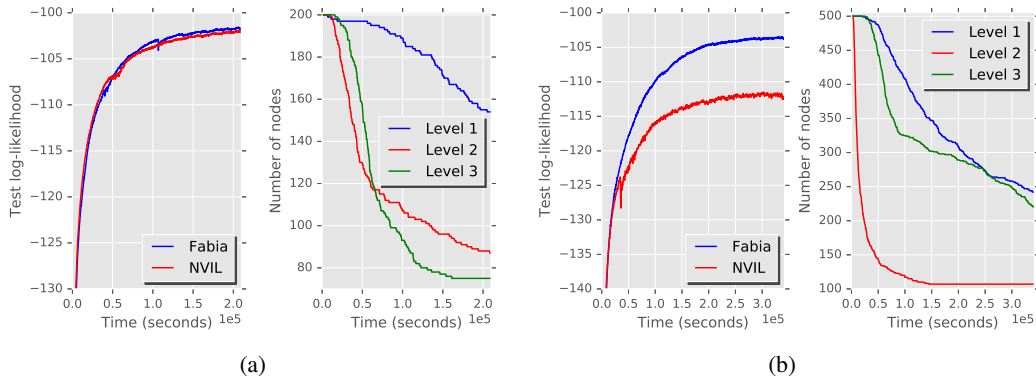

Figure 3: Test log-likelihood and the number of nodes in FABIA, as a function of CPU time on the MNIST dataset, for an SBN with initial size as (a) 200-200-200 (b) 500-500-500.

## 5.3 Topic Modeling

The two benchmarks we used for topic modeling are Reuters Corpus Volume I (RCV1) and Wikipedia, as in Gan et al. [10], Henao et al. [14]. RCV1 contains 794,414 training and 10,000 testing documents, with a vocabulary size of 10,000. Wikipedia is composed of 9,986,051 training documents, 1,000 test documents, and 7,702 words. The performance metric we use is the predictive perplexity on the test set, which cannot be directly evaluated. Instead, we follow the approach of $80/20\%$ split on the test set, with details provided in Gan et al. [10].

We compare FABIA against DPFA [10], deep Poisson factor modeling (DPFM) [14], MCEM [31], Over-RSM [33], and NVIL. For both FABIA and NVIL, we use a mini-batch of 200 documents. The results for other methods are cited from corresponding references. We test DPFA and DPFM with the publicly available code provided by the authors; however, no shrinkage of nodes are observed in our experiments.

Table 2 shows the perplexities of different algorithms on the RCV1 and Wikipedia datasets, respectively. Both FABIA and NVIL outperform other methods with marked margins.

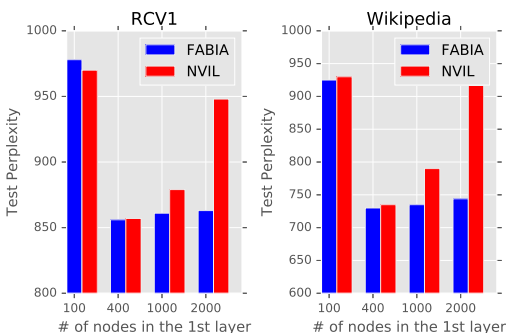

Figure 4: Test perplexities as a function of number of nodes in the first layer, in the two-layer case.

Interestingly, we note that FABIA does not shrink any nodes in the first layer, which is likely due to the fact that these two datasets have a large number of visible units and thus a sufficiently large first hidden layer is necessary. This requirement of a large first hidden layer to properly model the data may also explain why NVIL does not overfit on these datasets as much as it does on MNIST; the training set of these datasets being sufficiently large is another possible explanation. We also computed test time but did not observe any clear improvement of FABIA over NVIL, which may be explained by the fact that most of the computation is spent on the first layer in these two benchmarks.

In Figure 4, we vary the number of hidden units in the first layer and fix the number of nodes in other layers to be $400$. We use early stopping for NVIL to prevent it from overfitting with larger networks. For the networks with $100$ and $400$ nodes in the first layer, FABIA and NVIL have roughly the same perplexities. Once the number of nodes is increased to $1000$, FABIA starts to outperform NVIL with remarkable gaps, which implies that FABIA can handle the overfitting problem, as a consequence of its shrinkage mechanism for model selection. We also observed that setting a larger $\epsilon^{(1)}$ for the first layer in the 2000 units case for FABIA can stabilize its performance;

Table 2: Test perplexities and model size on the benchmarks. FABIA starts from a model initialized with 400 hidden units in each layer.

| | RCV1 | | Wikipedia | |
| | Size | Perplexity | Size | Perplexity |
|---|---|---|---|---|
| Over-RSM | 128 | 1060 | - | - |
| MCEM | 128 | 1023 | - | - |
| DPFA-SBN | 1024-512-256 | 964 | 1024-512-256 | 770 |
| DPFA-RBM | 128-64-32 | 920 | 128-64-32 | 942 |
| DPFM | 128-64 | 908 | 128-64 | 783 |
| NVIL | 400-400 | **857** | 400-400 | 735 |
| FABIA | 400-156 | **856** | 400-151 | **730** |

we choose this value by cross-validation. The results for three layers are similar and are included in Supplemental Materials.

## 6 Conclusion and Future Work

We develop an automatic method to select the number of hidden units in SBNs. The proposed gFIC criterion is proven to be statistically consistent with the model's marginal log-likelihood. By maximizing gFIC, the FABIA algorithm can simultaneously execute model selection and inference tasks. Furthermore, we show that FABIA is a flexible framework that can be combined with auto-encoding VB approaches. Our experiments on various datasets suggest that FABIA can effectively select a more-compact model and achieve better held-out performance. Our future work will be to extend FABIA to importance-sampling-based VAEs [5, 26, 24]. We also aim to explicitly select the number of layers in SBNs, and to tackle other popular deep models, such as CNNs and FNNs. Finally, investigating the effect of FABIA's shrinkage mechanism on the gradient noise is another interesting direction.

### Acknowledgements

The authors would like to thank Ricardo Henao for helpful discussions, and the anonymous reviewers for their insightful comments and suggestions. Part of this work was done during the internship of the first author at NEC Laboratories America, Cupertino, CA. This research was supported in part by ARO, DARPA, DOE, NGA, ONR, NSF, and the NEC Fellowship.

## Footnotes

[1] For derivation clarity, we assume only one hidden layer and drop the bias term in the SBN

[2]We list the number of nodes in the deeper layer first in all of our experiments.

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
