[Supplementary Material]

# Scalable Model Selection for Belief Networks
# Supplemental Materials

**Zhao Song[†], Yusuke Muraoka[∗], Ryohei Fujimaki[∗], Lawrence Carin[†]**

[†]Department of ECE, Duke University
Durham, NC 27708, USA
{zhao.song, lcarin}@duke.edu

[∗]NEC Data Science Research Laboratories
Cupertino, CA 95014, USA
{ymuraoka, rfujimaki}@nec-labs.com

## 1  Derivation of Minibatch gFIC in (7)

We first note that an unbiased estimator for (5) is

$$
\begin{aligned}
\widehat{\text{gFIC}_{\text{SBN}}} \;=\; & \max_q \mathbb{E}_q \Bigg[ -\frac{M}{2} \sum_j \ln \Big( \frac{N_{T+1}}{N_{mini}} \sum_{i=1}^{N_{mini}} h_{i+N_T, j} \Big) \\
& + \frac{N_{T+1}}{N_{mini}} \sum_{i=1}^{N_{mini}} \ln \frac{p(\boldsymbol{v}_{i+N_T}, \boldsymbol{h}_{i+N_T} | \widehat{\boldsymbol{\theta}})}{q(\boldsymbol{h}_{i+N_T} | \boldsymbol{\phi})} + \frac{MJ - D_{\boldsymbol{\theta}}}{2} \ln N_{T+1} \Bigg]
\end{aligned}
\tag{A1}
$$

Simplifying and ignoring constant terms in (A1) leads to $\widetilde{\text{gFIC}_{\text{SBN}}}$ in (7) of the main text.

## 2  Additional Results

Figure A1:  Test log-likelihood and the number of nodes in FABIA, as a function of CPU time on the MNIST dataset, for an SBN with initial size as (a) 200 (b) 500.

Figure A2: Test log-likelihood and the number of nodes in FABIA, as a function of the number of iterations on the MNIST dataset, for an SBN with initial size as (a) 200 (b) 500.

Figure A3: Test log-likelihood and the number of nodes in FABIA, as a function of CPU time on the MNIST dataset, for an SBN with initial size as (a) 200-200 (b) 500-500.

Figure A4: Test log-likelihood and the number of nodes in FABIA, as a function of the number of iterations on the MNIST dataset, for an SBN with initial size as (a) 200-200 (b) 500-500.

Figure A5: Test log-likelihood and the number of nodes in FABIA, as a function of the number of iterations on the MNIST dataset, for an SBN with initial size as (a) 200-200-200 (b) 500-500-500.

Figure A6: Test perplexities as a function of number of nodes in the first layer, in the three-layer case.

Table 1: Test perplexities and model size on the benchmarks, for NVIL and FABIA with three layers. FABIA starts from a model initialized with $400$ hidden units in each layer.

|  | RCV1 | | Wikipedia | |
|  | Size | Perplexity | Size | Perplexity |
| --- | --- | --- | --- | --- |
| NVIL | 400-400-400 | 859 | 400-400-400 | 731 |
| FABIA | 398-46-10 | 860 | 400-36-9 | 737 |