[Reviews · NeurIPS 2017]

Reviewer 1



The authors propose an approach for learning multi-layered sigmoid belief networks, where sparsity is enforced by imposing a regularizer on the number of nodes. In particular, they employ a variational lower bound on the marginal likelihood, and then applying a Laplace approximation. The resulting form includes a term that can be viewed as a regularizer on the model. The experiments show some improvements against other approaches, obtaining better likelihoods. On the plus side, the basic approach appears to be simple conceptually. The authors show some desirable properties under some assumptions. The synthetic experiments show that the approach seems to be doing reasonable things. On the real world experiments, the authors show that their approach achieves some improvements in terms of (bounds on the) test likelihood, but the improvements appear to be somewhat modest in this regard. The authors' approach also appears to learn more compact models. It was compared with other approaches that impose some sparsity, but it was stated several times that other approaches did not observe any shrinkage of nodes. It would be helpful here, if possible, to get a sense or intuition of why the authors' approach may be more successful at imposing sparsity here compared to other approaches. I also wonder how this work relates to work such as: "Learning the Structure of Deep Sparse Graphical Models. By R. Adams, H. Wallach, and Z. Ghahramani. In AISTATS, 2010. Here, they proposed the continuous Indian buffet process, that allows the number of nodes in each layer of a belief network to be learned, but also the depth. In contrast, the authors mostly cite prior works as being able to impose sparsity only on edges, and not on nodes.

Reviewer 2



The "FABIA" algorithm is used to maximize (a lower bound) on generalized factorized information criterion (gFIC). gFIC is a model score (here for SBNs). The presented method can be viewed as an approximate Bayesian approach where a variational approach is used for approximation of marginal log-likelihood. (Priors on model structure aren't considered, so we are implicitly using a uniform one.) gFIC is well-motivated and the FABIA approach to bounding it is sensible. The derivations in Section 3 constitute a strong part of the paper. The big issue is whether this method delivers the promised 4 benefits mentioned at the end of the abstract. The authors say of Fig 2(a) that: "In both one- and two-layer cases, the inferred model size is very close to the ground truth." but, in fact, what we see is that the *mean* size is close to the ground truth, the s.d. is reasonably large. 2(b) does show that the FABIA does better than the NVIL (when NVIL is not initialised with the correct size). On Table 1, there is a small win in terms of VLB, but the smaller size is some advantage. I'm not convinced that compactness per se is something people care about too much. No one looks at the SBN model structure to get 'insight'. These SBNs are used for predictions. On the other hand, as the authors note, a smaller SBN allows "faster testing", ie the SBN output is more rapidly computable. On topic modelling we again get small wins when comparing to NVIL. There is progress here. The issue is whether there is enough for a venue like NIPS. I think there is, but only just.

Reviewer 3



The authors propose a variational Bayes method for model selection in sigmoid belief networks. The method can eliminate nodes in the hidden layers of multilayer networks. The derivation of the criterion appears technically solid and a fair amount of experimental support for the good performance of the method is provided. I have to say I'm no expert in this area, and I hope other reviewers can comment on the level of novelty. detailed comments: - p. 1, l. 21: "Model selection is here the task of selecting the number of layers [...]": I got the impression that the proposed algorithm only eliminates individual nodes, not entire layers. Please clarify. - p. 2, ll. 76--77: Are there really no weights on the first layer? You forgot to define b. - p. 5, ll. 173--175: If nodes with expected proportion of 1's very small can be eliminated, why doesn't the same hold for nodes with expected proportion of 0's equally small? "When the expectation is not exact, such as in the top layers, [...]": Please clarify. How can we tell, when the expectation is not exact? And do you really mean 'exact' or just 'accurate', etc. What is the precise rule to decide this. - p. 6, ll. 226--227: Please explain what 10-5^2 and 25-15 means (is there a typo, or why would you write 5^2 instead of simply 25?). - p. 6, l. 239: "Our performance metric is the variational lower bound of the test log-likelihood." Why use a variational bound as a performance metric? Sounds like variational techniques are an approach to derive a criterion, but using them also to measure performance sounds questionable. Should the evaluation be based on a score that is independent of the chosen approach and the made assumptions? - references: please provide proper bibliographic information, not just "JMLR" or "NIPS". Add volume, page numbers, etc.